# Learning to Mutate with Hypergradient Guided Population

**Zhiqiang Tao**[1,4]*, **Yaliang Li**[2], **Bolin Ding**[2], **Ce Zhang**[3], **Jingren Zhou**[2], **Yun Fu**[4]
[1]Department of Computer Science and Engineering, Santa Clara University
[2]Alibaba Group
[3]Department of Computer Science, ETH Zürich
[4]Department of Electrical & Computer Engineering, Northeastern University
ztao@scu.edu, {yaliang.li, bolin.ding, jingren.zhou}@alibaba-inc.com
ce.zhang@inf.ethz.ch, yunfu@ece.neu.edu

## Abstract

Computing the gradient of model hyperparameters, *i.e.*, hypergradient, enables a promising and natural way to solve the hyperparameter optimization task. However, gradient-based methods could lead to suboptimal solutions due to the non-convex nature of optimization in a complex hyperparameter space. In this study, we propose a hyperparameter mutation (HPM) algorithm to explicitly consider a learnable trade-off between using global and local search, where we adopt a population of *student* models to simultaneously explore the hyperparameter space guided by hypergradient and leverage a *teacher* model to mutate the underperforming students by exploiting the top ones. The teacher model is implemented with an attention mechanism and is used to learn a mutation schedule for different hyperparameters on the fly. Empirical evidence on synthetic functions is provided to show that HPM outperforms hypergradient significantly. Experiments on two benchmark datasets are also conducted to validate the effectiveness of the proposed HPM algorithm for training deep neural networks compared with several strong baselines.

## 1 Introduction

Hyperparameter optimization (HPO) [4, 11] is one of the fundamental research problems in the field of automated machine learning. It aims to maximize the model performance by tuning model hyperparameters automatically, which could be achieved either by searching a fixed hyperparameter configuration setting [3, 22, 32, 9] from the predefined hyperparameter space or by learning a hyperparameter schedule along with the training process [17, 25]. Among existing methods, hypergradient [2, 26] forms a promising direction, as it naturally enables gradient descent on hyperparameters.

Hypergradient is usually defined as the gradient of a validation loss function w.r.t hyperparameters. Previous methods mainly focus on computing hypergradients by using reverse-mode differentiation [2, 6, 26], or designing a differentiable response function [12, 25] for hyperparameters, yet without explicitly considering the non-convex optimization nature in a complex hyperparameter space. Thus, while hypergradient methods could deliver highly-efficient local search solutions, they may easily get stuck in local minima and achieve suboptimal performance. This can be clearly observed on some synthetic functions which share a similar shape of parameter space to the HPO problem (see Sec. 4.1). It also leads to the question: *can we find a way to help hypergradient with global information?*

The population based hyperparameter search methods work as a good complementary to the hypergradient, such as evolutionary search [27, 5], particle swarm optimization [8], and the population based

training [17, 21, 14], which generally employ a population of agent models to search different hyperparameter configurations and update hyperparameters with a mutation operation. The population could provide sufficient diversity to globally explore hypergradients throughout the hyperparameter space. However, it is non-trivial to incorporate hypergradients in the population based methods due to a possible conflict between the hand-crafted mutation operation (*e.g.*, random perturbation) and the direction of hypergradient descent.

To address the above challenges, we propose a novel hyperparameter mutation (HPM) scheduling algorithm in this study, which adopts a population based training framework to explicitly learn a *trade-off* (*i.e.*, a mutation schedule) between using the hypergradient-guided local search and the mutation-driven global search. We develop the proposed framework by alternatively proceeding model training and hyperparameter mutation, where the former jointly optimizes model parameters and hyperparameters upon gradients, while the latter leverages a student-teaching schema for the exploration. Particularly, HPM treats the population as a group of student models and employs a teacher model to mutate the hyperparameters of underperforming students. We instantiate our teacher model as a neural network with attention mechanism and learn the mutation direction towards minimizing the validation loss. Benefiting from learning-to-mutate, the mutation is adaptively scheduled for the population based training with hypergradient.

In the experiments, we extensively discuss the properties of the proposed HPM algorithm and show that HPM significantly outperforms hypergradient and global search methods on synthetic functions. We also employ the HPM scheduler in training deep neural networks on two benchmark datasets, where experimental results validate the effectiveness of HPM compared with several strong baselines.

## 2 Related Work

Roughly we divide the existing HPO methods into two categories, namely, hyperparameter *configuration* search and hyperparameter *schedule* search. Hyperparameter configuration search methods assume that the optimal hyperparameter is a set of fixed values, whereas hyperparameter schedule search methods relax this assumption and allow hyperparameters to change in a single trail.

**Hyperparameter configuration**. For hyperparameter configuration search methods, we may divide existing methods into three subcategories: model-free, Bayesian optimization, and the gradient-based methods. The first subcategory includes grid search [31], random search [3], successive halving [18], Hyperband [22], etc. Grid search adopts an exhausting strategy to select hyperparameter configurations in pre-defined grids, and the random search method randomly selects hyperparameters from the configuration space with a given budget. Inspired by the amazing success of random search, successive halving [18] and Hyperband [22] are further designed with multi-arm bandit strategies to adjust the computation resource of each hyperparameter configuration upon their performance.

All the above HPO methods are model-free as they do not have any distribution assumption about the hyperparameters. Differently, Bayesian optimization methods [32, 16, 7]) assume the existence of a distribution about the model performance over the hyperparameter search space. This category of methods estimates the model performance distribution based on the tested hyperparameter configurations, and predicts the next hyperparameter configuration by maximizing an acquisition function. However, due to the distribution estimation, the computation cost of Bayesian optimization methods could be high, and thus the hyperparameter searching is time-consuming. Recently, BOHB [32, 9] utilizes model-free methods such as Hyperband to improve the efficiency of Bayesian optimization.

The gradient-based HPO method is closely related to this work. Pioneering works [2, 6] propose to employ the reverse-mode differentiation (RMD) to calculate hypergradients on the validation loss based on the minimizer given by a number of model training iterations. Following this line, research efforts [26] have been made to reduce the memory complexity of RMD to handle the large-scale HPO problem. A forward-mode differentiation algorithm is proposed in [12] to further improve the efficiency of computing hypergradients based on the chain rule and a dynamic system formulation.

**Hyperparameter Schedule**. Two representative ways of changing hyperparameters are gradient-based methods such as self-tuning networks (STN) [25] and mutation-based methods such as population based training (PBT) [17, 21, 14]. STN employs hypernetworks [24] as a response function to map hyperparameters to model parameters so that it could obtain hypergradient by backpropagating the validation error through the hypernetworks. PBT performs an evolutionary search over the

hyperparameter space with a population of agent models. It provides a discrete mutation schedule via random perturbation. The other two interesting works related to this regime include hypergradient descent [1] and online meta-optimization [35]. However, these two works both focus more on online learning rate adaptation rather than a generic HPO problem. The proposed HPM algorithm belongs to the category of hyperparameter schedule. Different from existing methods, HPM explicitly learns suitable mutations when optimizing hypergradient in a complex hyperparameter space.

# 3 Hyperparameter Mutation (HPM)

## 3.1 Preliminary

Given input space $\mathcal{X}$ and output space $\mathcal{Y}$, we define $f(\cdot; \theta, h) : \mathcal{X} \to \mathcal{Y}$ as a model parameterized by $\theta$ and $h$, where $\theta \in \mathbb{R}^D$ represents model parameters and $h \in \mathbb{R}^N$ vectorizes $N$ hyperparameters sampled from the hyperparameter configuration space $\mathcal{H} = \mathcal{H}_1 \times \cdots \times \mathcal{H}_N$. $\mathcal{H}_i$ is a set of configuration values for the $i$-th hyperparameter. Let $\mathcal{D}_{trn}, \mathcal{D}_{val} : \{(x, y)\}$ be the training and validation set. We define $\mathcal{L}(\theta, h) : \mathbb{R}^D \times \mathbb{R}^N \to \mathbb{R}$ as a function of parameter and hyperparameter by

$$\mathcal{L}(\theta, h) = \sum_{(x,y) \in \mathcal{D}} \ell(f(x; \theta, h), y), \tag{1}$$

where $\ell(\cdot, \cdot)$ denotes a loss function and $\mathcal{D}$ refers to $\mathcal{D}_{trn}$ or $\mathcal{D}_{val}$. Upon Eq. (1), we further define $\mathcal{L}_{trn}$ and $\mathcal{L}_{val}$ as the training and validation loss functions by computing $\mathcal{L}(\theta, h)$ on $\mathcal{D}_{trn}$ and $\mathcal{D}_{val}$, respectively. Generally, we train the model $f$ on $\mathcal{D}_{trn}$ with the fixed hyperparameter $h$ or a human-crafted schedule strategy, and peek at the model performance by $\mathcal{L}_{val}$ with the learned parameter $\theta$. Thus, the validation loss is usually bounded to the hyperparameter selection.

Hyperparameter optimization (HPO) solves the above issue, and it could be formulated as

$$\min_{h \in \mathcal{H}} \mathcal{L}_{val}(\theta^*, h) \text{ s.t. } \theta^* = \operatorname*{argmin}_{\theta} \mathcal{L}_{trn}(\theta, h), \tag{2}$$

which seeks for an optimal hyperparameter configuration $h^*$ or an optimal hyperparameter schedule. Hypergradient [2, 26, 30, 12] provides a natural way to solve Eq. (2) by performing gradient descent. However, due to the non-convex nature of a hyperparameter space, this kind of method may get stuck in local minima and thus lead to suboptimal performance. In contrast, the population based methods utilize a mutation-driven strategy to search the hyperparameter space thoroughly, which provides the potential to help hypergradient escape from local valleys. In this study, we focus on developing a *trade-off* solution between using hypergradient and the mutation-driven search.

## 3.2 Population Based Hyperparameter Search

We adopt a similar population based training framework as proposed in [17]. Let $\mathcal{S}_t = \{S_t^k\}_{k=1}^K$ be a population of agent models w.r.t $f(\cdot; \theta, h)$ at the $t$-th training step, where $S_t^k$ refers to the $k$-th agent model, $T$ represents the total training steps, and $K$ denotes the population size. Generally, the iterative optimization method (*e.g.,* stochastic gradient decent) is used to optimize model weights for each agent. Hence, for $\forall k$, one training step could be described as

$$\theta_{t+1}^k \leftarrow S_t^k(\theta_t^k, h_t^k), \tag{3}$$

where $S_t^k$ updates model parameters from $\theta_t^k$ to $\theta_{t+1}^k$ with a fixed hyperparameter $h_t^k$ during the training step. The population based hyperparameter search is given by

$$k^* = \operatorname*{argmin}_{k} \{\mathcal{L}_{val}(\theta_T^k, h_T^k)\}_{k=1}^K. \tag{4}$$

In Eq. (4), $\theta_T^k = S_{T-1}^k(S_{T-2}^k(\ldots S_0^k(\theta_0^k, h_0^k) \ldots, h_{T-2}^k), h_{T-1}^k)$ is obtained by chaining a sequence of update steps with Eq. (3) and the hyperparameters are updated through some pre-defined or rule-based mutation operations (*e.g.,* random perturbation). More specifically, we summarize the searching process with population based training [17] as follows.

- **_Train step_** updates $\theta_{t-1}^k$ to $\theta_t^k$ and evaluates the validation loss $\mathcal{L}_{val}(\theta_t^k, h_t^k)$ for each $k$. One training step could be one epoch or a fixed number of iterations. An agent model is *ready* to be exploited and explored after one step.

- **_Exploit_** $\mathcal{S}_t$ by selection methods, *e.g.*, the truncation selection, which divides $\mathcal{S}_t$ into three sets of *top*, *middle*, and *bottom* agents in terms of validation performance. The agent models in *bottom* exploit the *top* ones by cloning their model parameters and hyperparameters, *i.e.*, $(\theta_t^k, h_t^k) \leftarrow (\theta_t^*, h_t^*)$, where $k \in bottom$ and $*$ represents the index of a top performer.
- **_Explore_** the hyperparameters with a mutation operation, denoted as $\Phi$. As in [17], $\Phi$ keeps non-bottom agents unchanged, and randomly perturbs a bottom agent's hyperparameter.

The population based training (PBT) methods [17, 21] simultaneously explore the hyperparameter space with a group of agent models. PBT inherits the merits of random search and leverages exploit & explore strategy to alternatively optimize the model parameter $\theta$ (by training step) and hyperparameter $h$ (by mutation). This leads to a joint optimization over $\theta$ and $h$, and eventually provides an optimal hyperparameter schedule, *i.e.*, $h_0^{k^*}, \ldots, h_{T-1}^{k^*}$ given by Eq. (4), among the population of agents. However, PBT has two limitations. 1) For each training step, the joint optimization stays at a coarse level since $S_t(\theta_t, h_t)$ updates $\theta_t$ by fixing $h_t$. 2) The hyperparameters are mainly updated by the mutation operation, yet a learnable mutation is under-explored.

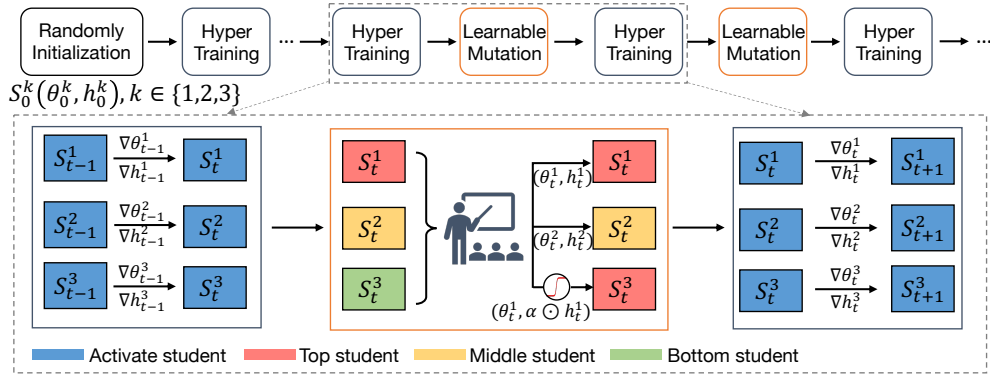

Figure 1: Illustration of the proposed HPM algorithm with an example of three student models. After performing one hypertraining step for all the students, HPM exploits the population by using the truncation selection and explores the cloned hyperparameter for the bottom student models with learnable mutations given by a teacher model.

### 3.3 Hypergradient Guided Population

We propose to use hypergradient to guide the population based hyperparameter search. To obtain hypergradient, we define $\theta(h) : \mathbb{R}^N \rightarrow \mathbb{R}^D$ as a response function of hyperparameter $h$ to approximate the model parameter $\theta$. By using $\theta(h)$, we could extend the agent model to $S_t(\theta_t(h_t), h_t)$, and formulate our hyperparameter mutation (HPM) scheduling algorithm as

$$\min_{h_T}\{\mathcal{L}_{val}(\theta_T^k(h_T^k), h_T^k)\}_{k=1}^K, \tag{5}$$

where $(\theta_T^k(h_T^k), h_T^k)$ is obtained by alternatively proceeding with one *hypertraining* step and one *learnable mutation* step as shown in Fig. 1. It is worth noting that, $h_T$ is optimized over the population in a sequential update way, *i.e.*, $(\theta_{t-1}^k(h_{t-1}^k), h_{t-1}^k) \rightarrow (\theta_t^k(h_t^k), h_t^k)$, where $h_t^k$ is updated by hypergradient and mutation at each step $t$. Thus, optimizing $h_T$ in Eq. (5) is equivalent to optimize the hyperparameter schedule: $h_0 \rightarrow \cdots h_t \cdots \rightarrow h_T$.

*Hypertraining* jointly optimizes $\theta$ and $h$ with hypergradients. Specifically, $(\theta, h)$ is updated by

$$\begin{aligned} \theta_t &= \theta_{t-1}(h_{t-1}) - \eta_\theta \nabla\theta, \\ h_t &= h_{t-1} - \eta_h \nabla h, \end{aligned} \tag{6}$$

where $\nabla\theta = \partial\mathcal{L}_{trn}/\partial\theta$ is the gradient of model parameter and $\nabla h$ is the hypergradient computed by

$$\nabla h = \frac{\partial\mathcal{L}_{val}(\theta(h), h)}{\partial\theta}\frac{\partial\theta}{\partial h} + \frac{\partial\mathcal{L}_{val}(\theta(h), h)}{\partial h}. \tag{7}$$

The computation of hypergradient in Eq. (7) is mainly depended on the response function $\theta(h)$. In this work, $\theta(h)$ is implemented by hypernetworks [24, 25], which provide a flexible and efficient way to compute hypergradients.

---
**Algorithm 1** Hyperparameter Optimization via HPM
---
Let $\mathcal{S}$ be a set of student models, and $T$ be the given budget
**for** $t = 1$ **to** $T$ **do**
$\quad$ **for** $S_{t-1}^k \in \mathcal{S}_{t-1}$ *(could be parallelized)* **do**
$\quad\quad$ Update $S_{t-1}^k(\theta_{t-1}^k, h_{t-1}^k)$ to $S_t^k(\theta_t^k, h_t^k)$ by one hypertraining step with Eq. (6) and Eq. (7)
$\quad$ Divide $\mathcal{S}_t$ into $top, middle, bottom$ students by the truncation section method
$\quad$ **for** $S_t^k \in bottom$ **do**
$\quad\quad$ Clone model parameters as $\theta_t^k \leftarrow \theta_t^*$ where $(\theta_t^*, h_t^*) \in top$
$\quad\quad$ Train the teacher network $g_\phi(h_t^k)$ with Eq. (10) conditioning on $(\theta_t^*, h_t^*)$
$\quad\quad$ Mutate the hyperparameter with Eq. (8) as $h_t^k \leftarrow g_\phi(h_t^k) \odot h_t^*$
**return** $\{h_0^*, \ldots, h_{T-1}^*\}, \theta_T^*$
---

*Learnable mutation* employs a similar exploit strategy as in Section 3.2 (without $h_t^k \leftarrow h_t^*$) and develops a student-teaching schema [10, 34] for exploration. Particularly, after updating $\mathcal{S}_{t-1}$ to $\mathcal{S}_t$ via one hypertraining step, we treat each agent $S_t^k \in \mathcal{S}_t$ as a student model and learn a teacher model to mutate the underperforming student's hyperparameters. The mutation module $\Phi$ is developed as

$$h_t^k = \Phi(h_t^k, h_t^*) = \alpha \odot h_t^*, \tag{8}$$

where $h_t^k \in bottom$, $h_t^* \in top$, $\odot$ is the hadamard product, and $\alpha \in \mathbb{R}^N$ denotes the mutation weights. In the following, we will show how to learn $\alpha$ with the teacher network.

### 3.4 Learning to Mutate

We formulate our teacher model $g_\phi$ as a neural network with attention mechanism parameterized by $\phi = \{W, V\}$, where $W \in \mathbb{R}^{N \times M}$, $V \in \mathbb{R}^{N \times M}$ are two learnable parameters and $M$ represents the number of attention units, as shown in Fig. 2. It takes input as a bottom student's hyperparameter $h_t^k$ and computes the mutation weights by

$$\alpha = g_\phi(h_t^k) = 1 + \tanh(c), c = W\text{softmax}(V^{\text{T}} h_t^k), \tag{9}$$

where $\alpha \in [0, 2]^N$ and $c \in \mathbb{R}^N$ is a mass vector that tries to characterize the mutation degree for each dimension of $h$. The benefits of using attention mechanism lie in two folds. 1) It provides sufficient model capability with a key-value architecture, which uses the key slots stored in $V$ to address different underperforming hyperparameters and assign the mutations with the corresponding memory slots in $W$. 2) $g_\phi$ enables a learnable way to adaptively mutate hyperparameters along with the training process, where $\alpha \to 1$ gives a mild mutation for a small exploration (update) step, and $\alpha \to 0$ or $\alpha \to 2$ encourages an aggressive exploration to the hyperparameter space.

We aim to learn the mutation direction towards minimizing $\mathcal{L}_{val}$. To this end, we train our teacher model $g_\phi$ conditioning on $(\theta_t^*, h_t^*)$ by

$$\min_{\phi=\{W,V\}} \mathcal{L}_{val}(\theta_t^*(h'), h_t'), \tag{10}$$

where $h_t' = \alpha \odot h_t^* = g_\phi(h_t^k) \odot h_t^*$. The param-eters of $g_\phi$ are updated by backpropagating the

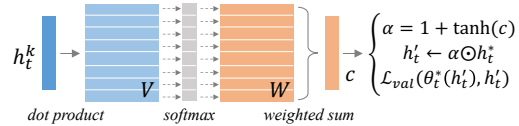

Figure 2: Illustration of our teacher model imple-mented by an attention network.

hypergradients given in Eq. (7) through the chain rule. By freezing the cloned model parameters and hyperparameters $(\theta_t^*, h_t^*)$, $g_\phi$ could be focused on learning the mutations to minimize $\mathcal{L}_{val}$. Please refer to the supplementary material for more details about training the teacher model.

Algorithm 1 summarizes the entire HPM scheduling algorithm. Particularly, HPM computes hypergra-dients with hypernetworks [24, 25], which add a linear transformation between hyperparameters and model parameters layer-wisely. The hypernetwork can be efficiently computed via feed-forward and backpropagation operations. Moreover, since the teacher network is trained with the frozen student model, the additional computing cost it brings in is much less than training a student model. Thus, the time complexity of HPM is mainly subject to the population size $K$. While the hypertraining step could be parallelized, the whole population cannot be asynchronously updated due to the centralized teaching process. This can be effectively addressed by introducing an asynchronous HPM, similar to [17]. We leave it as future work and focus on learning to mutate in this study.

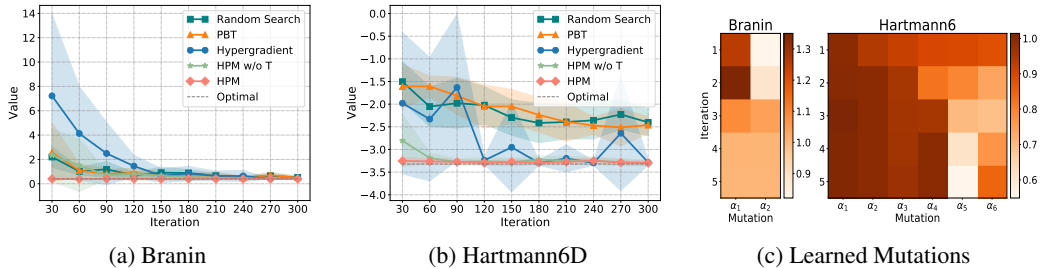

(a) Branin        (b) Hartmann6D        (c) Learned Mutations

Figure 3: Experiments on synthetic functions. (a)-(b) The mean performance computed by different methods along with the standard deviation over 10 trials, in terms of different given budget of iterations. (c) The average mutation values learned by HPM over 10 trials. In each trial, HPM runs 30 iterations in total with a population size of 5, resulting in 6 training steps and 5 mutations.

## 4 Experiments

### 4.1 Synthetic Functions

One common strategy for exploring the properties of hyperparameters is to perform hyperparameter optimization on synthetic loss functions [36]. These loss functions usually have many local minima and different shapes, and thus could well simulate the optimizing behavior of the real hyperparameters, yet work as much computationally cheaper testbeds than real-world datasets.

**Experimental Settings**. We employ the Branin and Hartmann6D function provided by the HPOlib[2] library, where Branin is defined in a two-dimensional space with three global minima ($f(h^*) = 0.39787$) and Hartmann6D is defined over a hypercube of $[0, 1]^6$ with one global minima ($f(h^*) = -3.32237$). We compare the proposed HPM with three baseline methods, including 1) random search [3], 2) population based training (PBT) [17], and 3) Hypergradient. We also compare HPM with HPM w/o T, which is the ablated HPM model without using a teacher network. It uses a random perturbation ($\alpha$ is randomly chosen from $[0.8, 1.2]$) for mutation instead. We ran the random search algorithm in HPOlib library and implement the PBT scheduler according to [17]. Note that, as we use the synthetic function $f$ to mimic the loss function of hyperparameters $h$, the hypergradient is directly given by $\partial f / \partial h$ and is optimized with the gradient descent algorithm.

**Hyperparameter Optimization Performance**. Fig. 3a and Fig. 3b compare the performance of different HPO methods on the Branin and Hartmann6D functions, respectively, where we have several interesting observations. 1) The hypergradient method generally performs better than the global search methods (*e.g.*, random search and PBT) on Hartmann6D rather than Branin, which is consistent with the fact that Hartmann6D has a less number of global minima than Branin. 2) There should be a trade-off between using hypergradient and global search methods (*e.g.*, PBT) according to their opposite performance on these two test functions. 3) The proposed teacher network leads to a more stable and faster convergence performance for HPM compared with HPM w/o T.

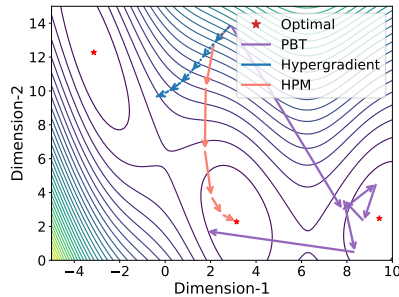

Figure 4: Illustration of the optimization behavior of PBT, Hypergradient, and HPM on the landscape of the Branin function. We run these three methods from the same initialization point with a budget of 30 iterations. HPM and PBT perform 6 updates as they both adopt a population size of 5.

**Mutation Schedule**. Fig. 4 shows the optimization steps of three methods on the Branin function, where we run PBT, hypergradient, and HPM from the same random initialization point with a budget of 30 iterations. As can be seen, the hypergradient decreases well along with the direction of gradient yet may get stuck in local minima. In contrast, while the PBT method could fully explore the hyperparameter space, it cannot achieve the global minimum without using the gradient guidance. Guided by the teacher network and hypergradient information, the proposed HPM moves towards the

Table 1: Performance comparison for the image classification task on the CIFAR10 dataset by validation/test loss and the language modeling task on the PTB corpus dataset by perplexity (PPL).

| Method | | CIFAR10 | | PTB | |
|---|---|---|---|---|---|
| | | Val Loss | Test Loss | Val PPL | Test PPL |
| Fixed | Grid Search | 0.7940 | 0.8090 | 97.32 | 94.58 |
| | Random Search | 0.9210 | 0.7520 | 84.81 | 81.86 |
| | Bayesian Optimization | 0.6360 | 0.6510 | 72.13 | 69.29 |
| | Hyperband [22] | 0.7156 | 0.7491 | 71.25 | 68.39 |
| Schedule | PBT [17] | 0.6253 | 0.6437 | 72.07 | 69.33 |
| | STN [25] | 0.5892 | 0.5878 | 71.49 | 68.29 |
| | HPM w/o T | 0.5724 | 0.5802 | 73.18 | 70.48 |
| | HPM | **0.5636** | **0.5649** | **70.49** | **67.88** |

global optimum adaptively, where HPM skips over several areas quickly and half steps to the end. Interestingly, this is consistent with the mutation schedule as shown in Fig. 3c on Branin, where HPM employs a larger mutation in the first three steps ($\alpha \to 0$ or $\alpha \to 2$) and mild mutations ($\alpha \to 1$) in the last two. Hence, benefiting from the learned mutation schedule, the proposed HPM is a good trade-off between using the hypergradient and mutation-driven update.

## 4.2 Benchmark Datasets

We validate the effectiveness of HPM for tuning hyperparameters of deep neural networks on two representative tasks, including image classification with CNN and language modeling with LSTM.

**Experimental Settings**. For a fair comparison to hypergradient, all the experiments in this section follow the same setting as in self-tuning networks [25], which is specifically designed for optimizing hyperparameters of deep neural networks with hypergradients. Particularly, we tune 15 hyperparameters, including 8 dropout rates and 7 data augmentation hyperparameters for AlexNet [20] in the CIFAR10 image dataset [19], and 7 RNN regularization hyperparameters [13, 33, 29] for LSTM [15] model in the Penn Treebank (PTB) [28] corpus dataset. We compare our approach with two groups of HPO methods as 1) *fixed hyperparameter* and 2) *hyperparameter schedule* methods. The first group tries to find a fixed hyperparameter configuration over the hyperparameter space, including grid search, random search, Bayesian Optimization[3] and Hyperband [22]. The second group learns a dynamical hyperparameter schedule along with the training process, such as population based training (PBT) [17] and self-tuning network (STN) [25]. Our HPM belongs to the second category.

**Implementation Details**. We implement PBT with different baseline networks (*e.g.*, AlexNet and LSTM) and use the truncation selection with random perturbation for exploitation and exploration according to [17]. For STN, we directly run the authors' code. We implement our HPM algorithm by using STN as a student model to proceed the hypertraining. HPM employs the same exploit strategy as in PBT and performs learnable mutation with a teacher model (*e.g.*, an attention neural network) for exploration. For both PBT and HPM, we take one training epoch as one *training step*, and do *exploit & explore* operation after each step. The teacher model in HPM is trained by one epoch on the validation set each time called by an underperforming student model. We also implement a strong baseline model as HPM w/o T, which incorporates hypergradient in the population based training without using a teacher network.

All the codes on benchmark datasets were implemented with Pytorch library. We set the population size as 20 and the truncation selection ratio as 20% for PBT, HPM w/o T, and HPM. We employed the recommended optimizers and learning rates for all the baseline networks and STN models following [25]. Our teacher network was implemented with 64 key slots and was trained with Adam optimizer with a learning rate of 0.001. For the fixed hyperparameter methods, we used the Hyperband [22] implementation provided in [23] and posted the results of the others reported in [25]. For all the hyperparameter schedule methods, we ran the experiments in the same computing environment. STN usually converges within 250 (150) epochs on the CIFAR-10 (PTB) dataset. Thus, we set $T$ as 250 and 150 for all the population based methods on CIFAR-10 and PTB, respectively.

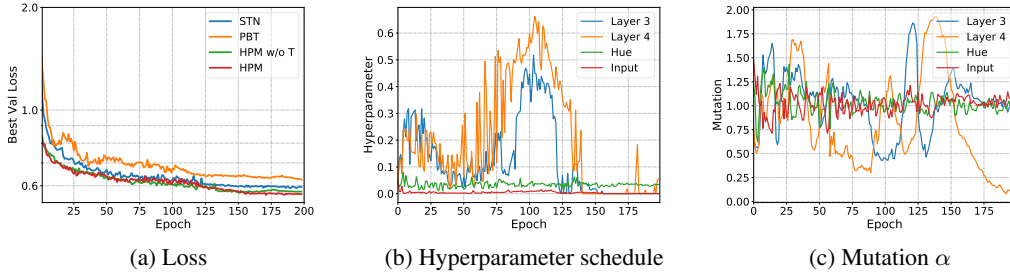

(a) Loss      (b) Hyperparameter schedule      (c) Mutation $\alpha$

Figure 5: Experiments on the CIFAR-10 image dataset. (a) The best validation loss of different methods over training epochs. (b) The learned schedule for 4 hyperparameters by HPM. (c) The mutation schedule given by HPM.

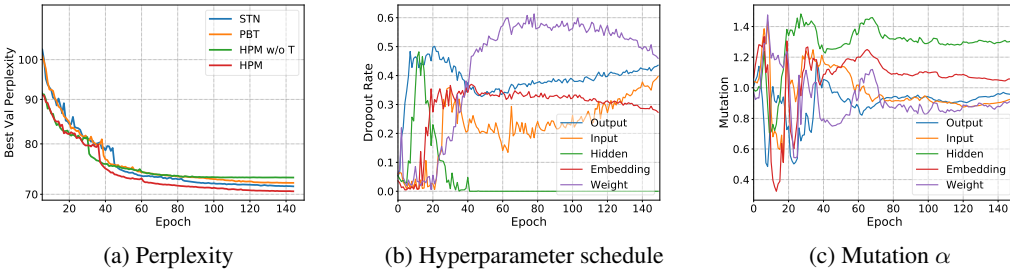

(a) Perplexity      (b) Hyperparameter schedule      (c) Mutation $\alpha$

Figure 6: Experiments on the PTB corpus dataset. (a) The best validation perplexity of different methods over training epochs. (b) The learned schedule for 5 dropout rates by HPM. (c) The mutation schedule given by HPM.

**Image Classification**. Table 1 reports the performance of the fixed hyperparameter and hyperparameter schedule methods on the CIFAR-10 dataset in terms of validation and test loss, respectively. As can be seen, the hyperparameter schedule methods generally perform better than the fixed ones and the proposed HPM scheduler achieves the best performance, which demonstrates the effectiveness of using HPM in tuning deep neural networks. Fig. 5a shows the best validation loss of different methods over training epochs, where the loss of HPM is consistently lower than PBT and STN. We also show the hyperparameter and mutation schedule learned by HPM in Fig. 5b and Fig. 5c. Specifically, we select four hyperparameters including the dropout rates of the input, the third and fourth layer activation, and the rate of adding noise on the hue of an image. We observe that the mutation has a consistent behavior with the hyperparameter. For example, HPM schedules the dropout rate of Layer 3 with a high variance at the early training stage and assigns it a stable small value after the 150-th epoch. Accordingly, the mutation $\alpha$ of Layer 3 oscillates between $[0.5, 1.75]$ before 150 epochs and then tends to be 1. For another example, as Hue and Input have a relatively stable schedule, their mutation weights spread around 1 with a small variance. These observations indicate that HPM can learn a meaningful mutation schedule during the training process.

**Language Modeling**. We summarize the validation and test perplexity of all the methods on the PTB corpus dataset in Table 1, where HPM also outperforms all the compared methods. One may note that HPM w/o T performs much worse than PBT and STN. This might be due to the conflict between hypergradient and the exploration of random perturbation, which justifies that HPM is not a trivial combination of PBT and STN, and supports that the proposed teacher network plays a key role in finding the mutation schedule. Fig. 6 shows the best validation perplexity of different methods over training epochs on the PTB dataset, as well as the hyperparameter and mutation schedules given by HPM, where a similar observation to the image classification experiment could be obtained.

## 4.3 Ablation Study

The proposed HPM method adopts a population-based training framework and learns the hyperparameter schedule by alternatively proceeding with the hypertraining and learnable mutation steps. To investigate the impact of different components in HPM, we provide more ablated models other than HPM w/o T as follows: 1) `RS+STN` combines STN [25] and random search (RS). We ran RS with the same given budget as the population size in HPM, *i.e.*, $K = 20$. 2) `HPM w/o H` freezes hyperparame-

Table 2: Ablation study on the CIFAR10 dataset by validation/test loss and the PTB corpus dataset by perplexity (PPL). We investigate the ablated models from four different aspects, including the hypertraining step, population-based training (PBT), learnable mutation, and teacher network.

| Methods | Model Components | | | | CIFAR10 | | PTB | |
|---------|---------------|-----|----------|---------|----------|-----------|---------|----------|
| | Hypertraining | PBT | Mutation | Teacher | Val Loss | Test Loss | Val PPL | Test PPL |
| RS + STN | ✓ | | | | 0.5817 | 0.5832 | 71.62 | 68.47 |
| HPM w/o H | | ✓ | ✓ | ✓ | 0.5944 | 0.6031 | 73.76 | 70.73 |
| HPM w/o M | ✓ | ✓ | | | 0.6139 | 0.6267 | 75.24 | 72.85 |
| HPM w/o T | ✓ | ✓ | ✓ | | 0.5724 | 0.5802 | 73.18 | 70.48 |
| HPM (T-MLP) | ✓ | ✓ | ✓ | ✓ | 0.5696 | 0.5745 | 70.91 | 67.94 |
| HPM [†] | ✓ | ✓ | ✓ | ✓ | **0.5636** | **0.5649** | **70.49** | **67.88** |

[†] indicates the full proposed model.

ters in the hypertraining step and only updates hyperparameters with learnable mutations. Thus, it could be treated as a PBT model with hypergradient-guided mutations. 3) HPM w/o M disables the mutation operation in HPM and, instead, performs one more hypergradient descent step on the cloned hyperparameters for the exploration purpose. 4) In HPM, the mutation is learned by a teacher model implemented with attention networks. Here HPM (T-MLP) employs a different implementation for the teacher model. Specifically, it implements the teacher model $g_\phi(h) = 1 + \tanh(W\sigma(V^\mathrm{T}h))$ by setting $\sigma$ as LeakyRelu rather than the softmax function in Eq. (9), in which case, it turns the attention networks as multilayer perceptron (MLP) networks.

Table 2 shows the ablation study results on two benchmark datasets, where our full model HPM consistently outperforms all the ablated models. On the one hand, RS+STN achieves a similar performance compared to STN [25], indicating that, without leveraging an effective *exploit & explore* strategy, a simple combination between local gradient and global search may not boost the performance significantly. On the other hand, while HPM w/o H adopts a learnable mutation, it only performs hypergradient descent with the teacher model, leading to hyperparameters will be updated slowly and cannot be seamlessly tuned along with model parameters. Hence, both hypertraining and learnable mutations are useful for optimizing hyperparameters.

We further compare HPM with two ablated models without using mutations (HPM w/o M) and the teacher network (HPM w/o T). Particularly, HPM w/o M degrades the performance due to over-optimizing hyperparameters and the lack of mutation-driven search; HPM w/o T underperforms since the potential conflict between hypergradient descent and the random-perturbation based mutation. Hence, the ablation studies in Table 2 demonstrate the effectiveness of learning mutations with a teacher model. Moreover, we also provide an alternative implementation of the teacher model with MLP networks, *i.e.*, HPM (T-MLP), which delivers comparative performance to the proposed HPM.

## 5 Conclusions

We proposed a novel hyperparameter mutation (HPM) algorithm for solving the hyperparameter optimization task, where we developed a hypergradient-guided population based training framework and designed a student-teaching schema to deliver adaptive mutations for the underperforming student models. We implemented a learning-to-mutate algorithm with the attention mechanism to learn a mutation schedule towards minimizing the validation loss, which provides a trade-off solution between using the hypergradient-guided local search and the mutation-driven global search. Experimental results on both synthetic and benchmark datasets clearly demonstrated the benefit of using the proposed HPM over hypergradient and the population based methods.

## Broader Impact

The proposed HPM algorithm addresses the challenge of combining local gradient and global search for solving the hyperparameter optimization problem. The proposed framework could be incorporated in many automated machine learning systems to provide an effective hyperparameter schedule solution. The outcome of this work will benefit both the academic and industry communities by liberating researchers from the tedious hyperparameter tuning work.

## Acknowledgments

We would like to thank the anonymous reviewers for their insightful comments and valuable suggestions. This work was supported by Alibaba DAMO Academy and the SMILE Lab (https://web.northeastern.edu/smilelab/) at Northeastern University.

## Footnotes

*Work done when Zhiqiang Tao interned at Alibaba Group and worked at Northeastern University.

[2]`https://github.com/automl/HPOlib`

[3] https://github.com/HIPS/Spearmint

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
