[Supplementary Material]

## Supplementary Material

We provide more details of training the teacher network in Section A, more experimental results on synthetic functions in Section B, and the hyperparameter settings for benchmark datasets in Section C.

## A  Details of Training Teacher Network

Different from the *exploit & explore* strategy in PBT [17], which directly clones $(\theta^*, h^*) \in top$ to an underperforming agent model $(\theta^k, h^k) \in bottom$ and mutates the cloned hyperparameter by random perturbation, we employ a teacher model $g_\phi$ to adaptively learn the mutation weights $\alpha$ for the exploration purpose, *i.e.*, $(\theta^k, h^k) \leftarrow (\theta^*, \alpha \odot h^*)$ where $\alpha = g_\phi(h^k)$. Here, we omit the iteration subscript $t$ for simplicity. The teacher network $g_\phi$ takes the *bottom* hyperparameter $h^k$ as input and outputs mutation weights $\alpha$ for exploring the local hyperparameter space centering on $h^*$. We formulate $g_\phi$ with Eq. (9) as $\alpha = g_\phi(h^k) = 1 + \tanh(W \operatorname{softmax}(V^{\mathrm{T}} h^k))$, and train $g_{\phi = \{W, V\}}$ by minimizing the validation loss given in Eq. (10).

Let $h' = \alpha \odot h^*$ denote the mutated hyperparameter during the training of $g_\phi$. To solve Eq. (10), we obtain the hypergradient regarding to $\alpha$ and backpropagate it to $\phi = \{W \in \mathbb{R}^{N \times M}, V \in \mathbb{R}^{N \times M}\}$. By substituting $h' = \alpha \odot h^*$ in Eq. (7), we obtain the hypergradient of $\alpha$ by

$$\nabla\alpha = \frac{\partial\mathcal{L}_{val}(\theta(h'), h')}{\partial\theta}\frac{\partial\theta}{\partial h'}\frac{\partial h'}{\partial\alpha} + \frac{\partial\mathcal{L}_{val}(\theta(h'), h')}{\partial h'}\frac{\partial h'}{\partial\alpha} = \nabla h|_{(\theta,h)=(\theta^*,h')} \odot h^*. \quad (11)$$

It is worth noting that, Eq. (11) is computed by fixing the model parameters and hyperparameters, $(\theta^*, h^*)$, of the top student model, so that the proposed HPM focuses on training the teacher network $g_\phi$ by learning the mutation weights to minimize the validation loss. To update the parameters of $g_\phi$, we compute the gradient of $\phi$ by employing the backpropagation algorithm with $\nabla\alpha$ as follows:

$$\begin{aligned}
\frac{\partial\mathcal{L}_{val}}{\partial W} &= \nabla\alpha(\sigma(V^{\mathrm{T}} h^k))^{\mathrm{T}}, \\
\frac{\partial\mathcal{L}_{val}}{\partial V} &= h^k \delta^{\mathrm{T}}, \delta = (W^{\mathrm{T}}\nabla\alpha) \odot \sigma'(V^{\mathrm{T}} h^k),
\end{aligned} \quad (12)$$

where $\sigma(\cdot)$ represents the softmax function and $\sigma'(\cdot)$ denotes its derivative. Thus, by using Eq. (11) and Eq. (12), we could train the teacher network by solving Eq. (10) with the SGD algorithm.

As shown in Algorithm 1, we train the teacher network one step when each time it is called by an underperforming student model, where the step refers to one iteration on synthetic functions and one epoch of the validation set on benchmark datasets in the experiment.

## B  Experiments on Synthetic Functions

In Section 4.1, we have shown the experimental results of HPM on two population synthetic functions, *i.e.*, the Branin and Hartmann6D functions. In the following, we will provide more details about synthetic functions and the implementation, as well as more results on the other two functions.

**Experimental Settings**. We used the Branin and Hartmann6D functions in Section 4.1. We show the details of these two functions as follows.

The Branin function is defined in a two-dimensional space of $x_1 \in [-5, 10]$ and $x_2 \in [0, 15]$. It is computed by

$$f(x) = (x_2 - \frac{5.1 x_1^2}{4\pi^2} + \frac{5 x_1}{\pi} - 6)^2 + 10(1 - \frac{1}{8\pi})\cos(x_1) + 10.$$

The Branin function has three global minima located at $[-\pi, 12.275]$, $[\pi, 2.275]$, and $[9.42478, 2.475]$, with a global optimal value of $f(x^*) = 0.397887$.

The Hartmann6D function is defined over a hypercube as

$$f(x) = -\sum_{i=1}^{4} \alpha_i \exp(-\sum_{j=1}^{6} A_{ij}(x_j - P_{ij})^2),$$

where $x_j \in (0, 1)$, $\alpha = [1.0, 1.2, 3.0, 3.2]^T$, $A$ and $P$ are two constant parameter matrices. The global minimum of the Hartmann6D function is $f(x^*) = -3.32237$ at $x^* = [0.20169, 0.150011, 0.476874, 0.275332, 0.311652, 0.6573]$.

We also conduct experiments on the other two population functions, including the Rosenbrock and the Bohachevsky functions. The Rosenbrock function is a valley-shaped function defined on the hypercube $x_i \in [-5, 10], i = 1, \ldots, d$. The Rosenbrock function is given by

$$f(x) = \sum_{i=1}^{d-1} [100(x_{i+1} - x_i^2)^2 + (x_i - 1)^2],$$

where we set $d = 2$ in our experiments. It has a global minimum value $f(x^*) = 0$ at $x^* = (1, \ldots, 1)$.

The Bohachevsky function is a bowl-shape function defined on the square $x_1, x_2 \in [-100, 100]$, which has a global minimum value $f(x^*) = 0$ at $x^* = (0, 0)$. In our experiment, we use the definition of the Bohachevsky function given by

$$f(x) = x_1^2 + 2x_2^2 - 0.3\cos(3\pi x_1) - 0.4\cos(4\pi x_2) + 0.7.$$

**Implementation Details**. In the experiment, we compare with random search, population based training (PBT), and hypergradient on synthetic functions. We adopt the random search algorithm in HPOlib library, and implement the PBT scheduler according to [17]. It is worth noting that, since the synthetic function only simulates the validation loss function (i.e., $\mathcal{L}_{val}$), it is in essence not suitable to the HPO methods that consider the model selection, like PBT. Hence, we use a simplified implementation here, where PBT performs one *training step* using random search and follow the same *exploit & explore* strategy in Sec. 3.2. The hypergradient is obtained by using the Stochastic Gradient Descent (SGD) algorithm with the synthetic function. We implement our HPM algorithm by treating a set of SGD optimizers as a population of student models, each of which proceeds the hypertraining as one gradient descent step. The same exploit strategy in PBT, *i.e.*, truncation selection [17], is used in our model. HPM w/o T is the ablated model of HPM without using the teacher network, which uses the random perturb (*i.e.*, $\alpha$ is randomly chosen from $[0.8, 1.2]$) instead and could be regarded as a hypergradient guided PBT scheduler.

All the codes on the synthetic functions were implemented with Autograd. The population size of PBT, HPM w/o T and HPM were set to be 5 and the ratio of truncation selection was set as 20%. The learning rate for all the SGD optimizers was 0.01, and we employed $M = 64$ keys for the attention mechanism in our teacher network.

(a) Rosenbrock

(b) Bohachevsky

Figure 7: Experimental results on the Rosenbrock (a) and Bohachevsky (b) functions.

**Experimental Results**. Fig. 7 compares the performance of different tuning methods on the Rosenbrock and Bohachevsky functions. Same to the Fig. 3 in Section 4.1, we show the mean performance of different methods and the standard deviation (std) over 10 trials. For a fair comparison, each method is given a certain budget of iterations, varied from 30 to 300 with a fixed increasing step of 30, and each iteration only evaluates the synthetic loss function once. As can be seen, the proposed HPM algorithm converges faster than the global search and gradient-based methods, which exhibits a similar observation result as in Fig. 3.

## C    Hyperparameters on Benchmark Datasets

We show the details of hyperparameters we tuned on the benchmark datasets as follows.

For image classification on the CIFAR-10 dataset, we study AlexNet [20] as the baseline model and employ a configuration space of 15 hyperparameters, including 8 dropout rates assigned on per-layer activations and input, and 7 data augmentation hyperparameters on controlling the noise added to hue, saturation, brightness, and contrast of an image, as well as the length and the number of cut-out holes. Following [25], all the dropout rates are ranged in $[0, 0.75]$; the noise ratios of saturation, brightness, and contrast are ranged in $[0, 1]$; the noise ratio of hue is ranged in $[0, 0.5]$; the number and the length of cut-out holes are in the ranges of $[0, 4]$ and $[0, 24]$, respectively.

For language modeling on the PTB corpus, we tune 7 RNN regularization hyperparameters for the LSTM [15] model, where we use a two-layer LSTM with 650 hidden units per layer and also set the word embedding size as 650. The tied weight is used for the embedding and softmax layer. Following [25], we adopt the hyperparameters as follows: 3 variational dropout [13] rates on the input, hidden states and output, 1 embedding dropout [13] rate, 1 DropConnect [33] rate on the recurrent hidden-to-hidden matrices, and 2 scaling coefficients for activation regularization and temporal activation regularization [29], respectively. All the dropout rates are ranged in $[0, 0.95]$, and the scaling coefficients of regularization are ranged in $[0, 4]$.