[Reviews · NeurIPS 2020]

Review 1

Summary and Contributions: This paper proposes a novel hypergradient-guided population-based training framework for hyperparameter optimization. The framework benefits utilizes hypergradients to jointly optimize the model and hyperparameters in a population of student networks while also using training a teacher network that learns to mutate the hyperparameters of student networks better via hypergradients from all the students. They show empirically that this performs better than hypergradient optimization or population-based search on several synthetic functions, and deep networks trained on CIFAR10(image classification) and PTB(Language Modeling).

Strengths: The work proposes a novel combination of hypergradient and population-guided search which to provide some benefits of both local-oriented hypergradient and global population-based search. They demonstrate the performance of the algorithm on both synthetic functions, and optimizing deep networks. They do ablation experiments with and without the teacher network learning the mutation operator and show that it improves performance. They include the code in the submission.

Weaknesses: It might benefit from an analysis of the additional cost of the computation of the hypergradients and whether optimization of the teacher network has significant impact on clock runtime time of the hyperparameter optimization. In addition, it may benefit from some more discussion of the variance in performance of the framework compared to other methods. _______________ AFTER REBUTTAL I thank the authors for the the author response which partially addressed my concerns and the additional ablation experiments make the paper significantly stronger especially the GB-HPO + RS baseline and the experiments showing the stability of the algorithm Unfortunately, after considering the other reviews, I believe that the paper is borderline partially due to concerns about the computational efficiency and wall clock time which were not sufficiently address in the rebuttal. While I agree updating the teacher network is significantly cheaper due to freezing the student model, in the current algorithm it must be done sequentially for half the population and training of the population is partially blocked by the teacher. The paper would benefit from significant analysis of the computational cost and parallelizability of the algorithms and wall time comparison with the baselines.

Correctness: The majority of the claims and method seems sound. However, it is unclear if the benchmark dataset results had multiple runs and they may benefit from additional trials to verify the stability of the algorithm.

Clarity: The paper is clear and well written.

Relation to Prior Work: The paper clearly discusses prior work and is well positioned.

Reproducibility: Yes

Additional Feedback: Might be useful to compare methods with this paper from the NAS space [1]. It also learned a mutation operator. [1] Chen, Yukang, et al. "Reinforced evolutionary neural architecture search." arXiv preprint arXiv:1808.00193 (2018).


Review 2

Summary and Contributions: This work explores the possibility of combining gradient-based (GB-HPO) and population-based hyperparameter optimization (PB-HPO). The authors propose a novel method, called Hypergradient Mutation (HPM) that builds on the work on self-tuning networks by MacKay et al. while adding a component of evolutionary search by keeping a population of student models and learning a mutation operation to change the hyperparameter values of underperforming models. The authors present experiments on test functions and two benchmark datasets to demonstrate the effectiveness of their approach.

Strengths: The main strengths of the paper lie in its novelty: - The idea of combining a local search strategy with a global one in the context of HPO seems a natural direction; and, to the best of my knowledge, this is the first work that attempts to do so with GB-HPO and PB-HPO. - Also learning mutation operations by gradient descent is a novel and interesting research direction.

Weaknesses: My main concern is about clarity as detailed below. Unfortunately I do not think this issue is limited to poor exposition, but rather impacts also the quality of the presentation and derivations necessary to properly understand the method proposed. Furthermore - on the experimental side, while the authors provide comparison with other method, I find it difficult to put the results in perspective ((*) could you please report also accuracy?); also the choice of datasets, but especially neural models, is rather limited. Regarding the method: - (*) There are quite a number of components at work, and I don't think the reader is in the position to judge the effectiveness of each of them with the material presented in this work. While I appreciate the presented ablation, I would really like to see two other very natural ways of proceeding: 1) GB-HPO with a global search strategy, like random search (essentially a multi-start method in the context of HPO) (2) Only learning to mutate (without ``hyper-training'').

Correctness: Claims: For me it is far from obvious how the method would adapt to other ways to compute the hypergradient (e.g. with RMD). I think that sentence should be either clarified (maybe in the appendix) or removed. Empirical evaluation: not sure how fair are the experiments in the final section, specifically regarding runtime. (*) Does HPM take approximately 20 times STN (since it effectively trains 20 models)?

Clarity: I think the mathematical exposition should be improved. Specifically - Eq. (1) clashes with subsequent definitions (suggestion: define L(theta, x, D) = ... and then L_val = L(theta, x, D_val), etc.) - (*) Eq (3): it is not clear what S is. Is it a model (student?) or an update rule (SGD?) or something else? - (*) Eqs. (4) and (5) Why subscript T? In (5) Does HPM optimizes only for the last step? - The authors should try, if possible, to give a ``final'' view of the algorithm in terms of an optimization problem (or, perhaps, as a sequence of optimization problems?). In the end, what are the variables being optimized? While the paper is not particularly difficult to follow ``per se'', I think there are numerous passages and expression choices that need attention. This really hinders the clarity of the paper. I'll give a non-exhaustive list: - L20 hypergradient (I would say it's an object, not a method) - L29 - L96 sampled? - L106-108 (clash between``solve'' and ``stuck``) - L125-126 - L136 (optimal? In which sense?) - L139 (why coarse?) - L180 (long-term behaviour?) - L196-197: the hypergradient is a gradient in this case, so the method you are confronting to is simply gradient descent (and I don't quite think it's stochastic in such a simple setting; where would the noise come from?). This should be clearly stated.

Relation to Prior Work: Partially; the related work section could be better curated. I found some sentences misleading; e.g. about limitations of Bayesian optimization and work of F. Pedregosa which actually does not involve RMD. Also, since I think the method relies so much on STN, I would have appreciated a short explanation with some details in the background section to better appreciate the differences with HPM. The STN approach to HPO and is quite different to the other works in GB-HPO, and this is not properly explained in the paper.

Reproducibility: Yes

Additional Feedback: I marked with (*) the points that I find particularly relevant. This is a second iteration of the work that I am reviewing, and I appreciate the effort of the authors in improving the manuscript. Nevertheless, I believe that it still requires polishing both on an exposition point of view, and also in terms of the method proposed and the experimental side (especially missing the two ablations I described above). Thus, I believe it is not ready for publication yet. __________ AFTER REBUTTAL I thank the authors for their reply and clarification. I appreciate the additional experimental result presented and therefore decided to rise my score to 5. I believe the paper would benefit from further editing, especially about notation and related work section. Furthermore, accuracy scores for CIFAR10 seem very far from modern baselines, which may cast some doubt on the benefit of using the method on more realistic applications (although I understand that the choice could have been dictated by easier comparison with previous work). For these reasons I still cannot advise acceptance at this stage.


Review 3

Summary and Contributions: This paper mainly introduces the use of a group of agent models to search for different configuration hyperparameters, and update the hyperparameters by mutation operation. The author thinks: if we can consider the direction of hypergradient when mutating, we can not only avoid conflicts between the direction of hand craft mutation operation and that of gradient descent, but also use global information to do hypergradient. The idea is natural and interesting.

Strengths: -- This paper combines the two methods of hypergradient optimization and population based optimization, which is a certain degree of innovative. -- The experimental part specially shows the trajectories of different optimization methods, which well proves the viewpoint of this paper.

Weaknesses: -- In this paper, the parameters of the attention mechanism network need to be retrained every time, which is time-consuming. -- Tanh and softmax functions are used in the g_{\phi}(h_t^k) network, but there is no comparative experiment on why to choose these two functions and structures. -- Why is it that replacing the parameter of mutation with a network can be regarded as hypergradient directed mutation? Please give a more specific explanation. -- Due to the lack of experiments on large-scale Imagenet data set, it is necessary to supplement to prove the effectiveness of the method.

Correctness: Hope to explain the Weaknesses mentioned above.

Clarity: Ditto.

Relation to Prior Work: Yes, clearly.

Reproducibility: Yes

Additional Feedback:


Review 4

Summary and Contributions: This work proposes the hyperparameter mutation (HPM) algorithm to combine the benefits of global search like PBT and the local search like hypergradient. Specifically it uses a population of student models like in PBT and interleaves a hyper-training step that uses hypergradient and a learnable mutation step that clones and mutates the top students. Additionally, the mutations are guided by a "teacher" model that learns to generate better mutations through hypergradients on the validation set. The proposed method is evaluated on the synthetic functions and tuning hyperparameters for deep neural network on CIFAR-10 and PTB, and performed better than baselines like PBT and STN.

Strengths: (1) The proposed method (HPM) that combines PBT and hypergradient is intuitive and well motivated and performs better than both PBT and hypergradient methods. (2) The ablation study (HPM w/o T) showed the benefits of the learnable mutations. (3) The evaluation is performed on both synthetic and real benchmarks.

Weaknesses: (1) The reason behind the gain from learnable mutations is a bit unclear. From algorithm 1 and equation 10, it seems the teacher network is trained on a given h^{k}_t before computing the mutation over it. So the mutation is guided by the hypergradient. If that's the main reason, perhaps you don't even need the teacher network and learnable mutations. Instead you just need to add one more hypergradient update step over the hyperparameters after cloning the top student models. Some simpler baseline like this should be compared with to justify introducing additional complexity of a teacher network. Another minor issue is the concern on the fairness of comparison with other methods, since the teacher network training also requires computation, which should be counted as part of the budget used in HPM. (2) The exact form of the teacher model is not very well motivated and justified. Attention mechanism is usually applied in situations where you use a query to attend to a number of items, for example, using a query word to attend to a number of other words in a sentence. However, there aren't any other items to attend to in this case and W and V are all just parameter matrices. It would be helpful to compare against some simpler forms, for example, just a multilayer feedforward networks, to justify the advantage of using attention mechanism here. ==================== Thanks for the author response, which addressed some of my concerns. I have increased my score accordingly.

Correctness: The evaluation is correct and supports the advantage of HPM.

Clarity: The paper is mostly well written and easy to follow.

Relation to Prior Work: This work is compared with other hyperparameter optimized methods, especially the population and hypergradient based methods.

Reproducibility: Yes

Additional Feedback:

[Author Response · NeurIPS 2020]

**Reviewer 1:** *1. Computation analysis.* Thanks for this useful suggestion! The proposed HPM computes the hyper-gradient with hypernetworks following STN [8], which adds a linear transformation between hyperparameters and model parameters layer-wisely. Thus, the additional computing cost is comparable to the original one and the whole model could be also efficiently trained by feed-forward and backpropagation operations. On another hand, we train the proposed teacher network (*i.e.*, a small attention network) by freezing the student model, leading to a computational cost less than one hypergradient descent step. A more detailed computational analysis will be added in the final version.

*2. Performance variance & Reference.* We ran HPM on CIFAR-10 three times and obtained validation/test loss as 1) 0.5598/0.5664; 2) 0.5606/0.5640; and 3) 0.5647/0.5704, showing our result is relatively stable. This is consistent with the observations on synthetic functions. We also thank the reviewer for pointing out the valuable reference. Due to the limited time and space here, we will report performance variance and discuss the provided reference in the final version.

**Reviewer 2:** *1. Experimental setting.* Thanks for the useful suggestion! In this work, we mainly follow the experimental setting in [8] for a fair comparison to other HPO methods. The validation/test accuracy (%) of PBT [7], STN [8] and HPM on CIFAR-10 are 78.5/78.1, 80.3/80.1, and 81.7/81.1, respectively. We will include them in the final version.

*2. More baseline results.* In Table 1, we implement 1) GB-HPO + RS by running STN [8] with Random Search given 20 trials and 2) HPM w/o hypertraining by only updating hyperparameters with learnable mutation. Compared with HPM, GB-HPO + RS may not fully explore the hyperparameter space due to the lack of mutation-driven search. While the HPM w/o hypertraining adopts learnable mutation, the hypergradient will decrease slowly and the hyperparameters cannot be seamlessly updated along with model parameters. More results will be included in the final version.

*3. Further questions.* 1) $S$ in Eq (3). $S$ denotes a agent model in the population-based training, which maintains its parameters $(\theta, h)$ and performs one training step (with SGD) once being called. 2) Subscript $T$ in Eqs (4-5). $h_T$ is obtained in a chained update sequence, $(\theta_t^k(h_t^k), h_t^k) \leftarrow (\theta_{t-1}^k(h_{t-1}^k), h_{t-1}^k)$, where $h_t$ is updated by hypergradient and mutation in each step $t$. Thus, minimizing $h_T$ is equivalent to minimize this schedule: $h_T \leftarrow \cdots h_t \cdots \leftarrow h_0$. We will clarify these in the final version.

Table 1: More baseline results on CIFAR-10.

| Methods | Val Loss | Test Loss |
|---|---|---|
| GB-HPO + RS | 0.5817 | 0.5832 |
| HPM w/o hypertraining | 0.5944 | 0.6031 |
| HPM (proposed method) | **0.5636** | **0.5649** |

**Reviewer 3:** *1. Additional cost by attention networks.* Thanks for the valuable feedback! The proposed teacher network (*i.e.*, attention networks) is retrained for adapting to the model training process and mutating the hyperparameters on the fly. We agree that it will bring additional cost. Fortunately, the teacher network is trained on the validation set by freezing the student model, which needs a much less computing cost than training students.

*2. Activation functions.* Previous works like PBT [7] mainly use discrete mutation weights sampled from $\{0.8, 1.2\}$. To empower the flexibility of mutation, we leverage the tanh function to describe the mutation degree in $[-1, 1]$, leading to continuous mutation weights in $[0, 2]$ with Eq. (9). The softmax function is used to compute attention scores. Table 2 compares using LeakyRelu and Softmax in teacher model. We will provide more comparison results in the final version.

*3. Hypergradient directed mutation.* Thanks for the useful suggestion! We train the teacher model by minimizing $\mathcal{L}_{val}$ w.r.t the mutated hyperparameters. Thus, the hypergradient could be backpropagated to mutation weights and update the teacher network. Due to the limited space, please refer to Appendix A in the supplementary material for more details.

*4. Large-scale experiments.* The hypernetworks inside HPM are scalable and memory-efficient to compute hypergradients. By using the population-based training, HPM could be further parallelized to handle large-scale datasets.

**Reviewer 4:** *1. Learnable mutation.* Thanks for this useful suggestion! The teacher model is trained along with hypergradient descent to mutate hyperparameters adaptively, which could provide aggressive mutations in early training steps (when $h_t^k$ exhibits a high variance) and tend to mild mutations when $\mathcal{L}_{val}$ gets converged (see Fig. 4 in the paper). We implement HPM w/o learnable mutation by performing one more hypergradient update step over the cloned hyperparameters. As shown in Table 2, this baseline method degrades the performance due to over-optimizing hyperparameters (the cloned model parameters remained unchanged) and the lack of mutation-driven search.

*2. Implementation of teacher model.* We thank the reviewer for this great suggestion! In our paper, we implement the teacher model as attention networks, *i.e.*, $g_\phi(h) = 1 + \tanh(W\sigma(V^{\mathrm{T}}h))$ where $\sigma$ denotes the Softmax function. We expect to use attention mechanism to make $V$ memorize different hyperparameter queries and $W$ focus on learning mutation degree. However, the main contribution of HPM is to learn the mutations with a teacher model for combining the local hypergradient and global population-based search. Hence, some other common network choices in the learning-to-learn regime, like MLP, can also be used as the teacher model of HPM. Particularly, we could implement teacher-MLP by setting the activation function $\sigma$ in $g_\phi$ other than Softmax, *e.g.*, setting $\sigma$ as LeakyRelu. Table 2 shows the comparison result between these two teacher forms.

Table 2: More ablation studies on CIFAR-10.

| Methods | Val Loss | Test Loss |
|---|---|---|
| HPM w/o learnable mutation | 0.6139 | 0.6267 |
| HPM (the proposed method) | **0.5636** | **0.5649** |
| HPM (T-MLP-LeakyRelu) | 0.5696 | 0.5745 |

[Meta-Review · NeurIPS 2020]

This paper got mixed reviews initially. On the positive side, most of the reviewers agree the idea of this paper is interesting and novel. The results are good. However, on the negative side, they also share concerns on the potentially high computation complexity of the proposed method and clarity on the presentation of this paper. In the author's response, they provide additional experiments that indeed help and two reviewers increase their ratings. However, the concerns on the computation cost and presentation still remain. Reviewers also exchange their opinion in the discussion phase. AC reads the paper and would like to encourage to explore novel idea. The potentially high cost would be an issue of the proposed method but seems addressable in the future. Based on this consideration, AC recommends acceptance. Authors should include their promised revision into the final version and improve the presentation quality.